# Cytokine Signatures in Psoriatic Arthritis Patients Indicate Different Phenotypic Traits Comparing Responders and Non-Responders of IL-17A and TNFα Inhibitors

**DOI:** 10.3390/ijms24076343

**Published:** 2023-03-28

**Authors:** Marie Skougaard, Sisse Bolm Ditlev, Magnus Friis Søndergaard, Lars Erik Kristensen

**Affiliations:** 1The Parker Institute, Bispebjerg and Frederiksberg Hospital, Nordre Fasanvej 57, 2000 Frederiksberg, Denmark; 2Copenhagen Center for Translational Research, Bispebjerg and Frederiksberg Hospital, Bispebjerg Bakke 23, 2400 Copenhagen, Denmark; 3Department of Clinical Immunology, Aarhus University Hospital, Palle Juul-Jensens Boulevard 99, 8200 Aarhus, Denmark; 4Department of Clinical Medicine, Faculty of Health and Medical Sciences, University of Copenhagen, Blegdamsvej 3b, 2200 Copenhagen, Denmark

**Keywords:** psoriatic arthritis, cytokines, bDMARDs, phenotypes, principal component analysis

## Abstract

This study aimed to explore the dynamic interactions between 32 cytokines and biomarkers in Psoriatic Arthritis (PsA) patients to compare cytokine signatures of treatment responders and non-responders. Biomarkers were measured before and after four months of treatment in 39 PsA patients initiating either Tumor Necrosis Factor alpha inhibitor (TNFi) or Interleukin-17A inhibitor (IL-17Ai). Response to treatment was defined by the composite measure, Disease Activity in Psoriatic Arthritis (DAPSA). A two-component principal component analysis (PCA) was implemented to describe cytokine signatures comparing DAPSA50 responders and non-responders. The cytokine signature of TNFi responders was driven by the correlated cytokines interferon γ (IFNγ) and IL-6, additionally associated with IL-12/IL-23p40, TNFα, and CRP, while the cytokine signature of TNFi non-responders was driven by the correlated cytokines IL-15, IL-8, and IFNγ. IL-17Ai responders were characterized by contributions of strongly correlated Th17 inflammatory cytokines, IL-17A, IL-12/IL-23p40, IL-22 to the cytokine signature, whereas IL-17A and IL-12/IL-23p40 did not demonstrate significant contribution in IL-17Ai non-responders. Based on PCA results it was possible to differentiate DAPSA50 responders and non-responders to treatment, endorsing additional examination of cytokine interaction models in PsA patients and supporting further PsA patient immune stratification to improve individualized treatment of PsA patients.

## 1. Introduction

Psoriatic Arthritis (PsA) is an inflammatory disease mediated by interacting innate and adaptive immune mechanisms [1,2], causing known symptoms including arthritis, cutaneous and nail psoriasis, dactylitis, and enthesitis [3]. However, the immunopathogenesis of PsA remains unclear [4].

Existing knowledge shows that stimulation of the innate immune response induces secretion of pro-inflammatory cytokines, including tumor necrosis factors-α (TNFα), Interleukin (IL)-6 IL-12, IL-22, and IL-23 from neutrophils, dendritic cells, monocytes, and macrophages in inflamed tissues [5]. Tissue inflammation leads to an upregulation of tissue-homing chemokines promoting the recruitment of additional inflammatory immune cell subtypes [6,7] activated by antigen-presenting cells [8]. The increased production and release of pro-inflammatory cytokines such as TNFα, interferon-γ (IFNγ), and IL-12/IL-23 [9] stimulate the activation and differentiation of the inflammatory T cell subset; T helper cell type 1 (Th1) and T helper cell type 17 (Th17) [10]. Th1 cells and Th17 cells are both considered to have a pathological effect in PsA through the secretion of IFNγ, IL-17, IL-22, etc. amplifying and sustaining inflammation in PsA [5]. The interactions between cells and mediators of the innate and the adaptive immune system inducing, amplifying, and sustaining inflammation in PsA are a complex and poorly understood process. Thus, researchers have been able to target treatment towards the immune systems to relieve PsA patients of many debilitating symptoms [11].

Medical therapies targeting the inflammatory pathway by the inhibitory effect on the inflammatory pathway of PsA include TNFα inhibitors (TNFi), IL-17A inhibitors (IL-17Ai) and IL-17A/F inhibitors, IL-12/IL-23p40 inhibitors, IL-23 inhibitors, and Janus kinase inhibitors (JAKi) [12]. The therapies have shown a tremendous impact treating PsA [13,14] significantly improving the degree of arthritis, enthesitis, and psoriasis, etc. [15]. However, cytokine inhibition still exhibits an incomplete effect in some patients [16], including side-effects such as TNFi-induced psoriasis in a subgroup of patients [17]. The diversity in clinical effects of available medical therapies demonstrates the need for a better understanding of the PsA immunopathogenesis and improved stratification of PsA patients prior to initiation of treatment.

Here we present follow-up data examining cytokine signatures of the immune response and inflammation associated biomarkers and cytokines in PsA patients initiating cytokine inhibition therapies, either TNFi or IL-17Ai due to active PsA. The primary objective of the study was to examine the interplay between cytokines associated with PsA immunopathogenesis with a two-component principal component analysis (PCA), i.e., describe cytokine signatures at baseline in responders versus non-responders of TNFi or IL-17Ai, aiming to explore differences in the immune response, that might explain the various effects of cytokine inhibition therapies. The secondary objective was to evaluate change to the cytokine signatures after four months of treatment.

## 2. Results

### 2.1. Baseline Characteristics

Paired baseline and four-months follow-up data, including clinical and biomarker data, were available from 39 PsA patients initiating biological disease modifying anti-rheumatic drugs (bDMARDs); 11 TNFi responders, 9 TNFi non-responders, 7 IL-17Ai responders, and 12 IL-17Ai non-responders. Additionally, data on 10 PsA patients initiating the conventional synthetic disease modifying the anti-rheumatic drug (csDMARD) methotrexate (MTX) were available for ancillary analysis to compare baseline cytokine signatures of the bDMARD initiators with a baseline cytokine signature of a patient group that was close to treatment naïve (Table 1), i.e., describing the inflammatory response of PsA. 

There were no differences between (1) responders versus non-responders to TNFi and IL-17Ai and (2) PsA patients initiating either TNFi, IL-17Ai or MTX at baseline comparing sex, age, disease duration, DAPSA, swollen and tender joint count (SJC and TJC), SPARCC enthesitis score, ultrasound (US) score (synovial hypertrophy), and Psoriatic Area Severity Index (PASI) (Table 1). A tendency (*p* = 0.049) indicated that PsA patients initiating TNFi had been treated with more bDMARDs prior to initiating the current TNFi than PsA patients initiating IL-17Ai and MTX (Table 1). This is considered cohort specific and most likely due to Danish treatment guidelines, meaning that TNFi is most often given as first line bDMARD in treatment of PsA. No differences in baseline biomarker levels were found comparing TNFi responders and non-responders, while IL-17Ai responders presented with higher VEGF-A (*p* = 0.005), VEGF-C (*p* = 0.022), MIP-1β (*p* = 0.002), TARC (*p* = 0.010), IL-7 (*p* = 0.017), IL-10 (*p* = 0.045), IL-8 (0.004), MIP-3α (*p* = 0.036), and CRP (*p* = 0.009) (Appendix A). Further presentation of results and discussion will primarily focus on the interaction between evaluated PsA associated cytokines. 

### 2.2. Cytokine Signatures at Baseline in PsA Patients Initiating TNFi

For TNFi responders, eigenvalues demonstrated 29.0% and 17.7% of the variance explained within component 1 (C1) and component 2 (C2), respectively. Biomarkers of C1 with correlations >0.500 included strongly correlated inflammatory cytokines IFNγ (C1 loading [C1L] 0.761) and IL-27 (C1L 0.783) suggesting the influence of Th1-mediated immune response mechanisms in the cytokine signature of TNFi responders [18] (Table 2). This is further supported by strongly correlated chemokine IP-10/CXCL10 (C1L 0.867) secreted in response to IFNγ during inflammation [19]. The suggested importance of Th1 associated inflammation is confused by the significant contribution to C1 from the polyfunctional cytokine IL-6 (C1L 0.712) known to suppress Th1 differentiation [20] and promote Th17 polarization [21]. However, strongly correlated cytokines of C2; IL-6 (C2L 0.559) and concomitant TNFα (C2L 0.555), IL-8 (C2L 0.846), IFNγ (C2L 0.531), IL-7 (0.609), IL-16 (0.592), and IL-12/IL-23 (C2L 0.663) support the inflammatory properties of IL-6 and the presence of an active inflammatory process (Table 2). Interestingly, no significant contribution from IL-17A was demonstrated to either C1 (C1L −0.029) or C2 (C2L 0.455). Overall, the PCA correlation plot (Figure 1) demonstrated a contribution >5% to C1 or C2 from the above-mentioned associated cytokines, including IFNγ, IP-10, and IL-6 within the same quadrant positively correlated to C1 and C2. These cytokines were additionally associated to correlated cytokines (a) IL-27 and IL-22, and (b) TNFα, CRP, and IL-8 on the opposite side. Cytokines of (a) and (b) were inversely correlated. In the dataset reporting TNFi non-responder signatures, C1 explained the majority of the variance with 39.6%, while C2 explained only little variance with 14.4%. The cytokine signature of TNFi non-responders implied heterogenous mechanisms involving strongly correlated cytokines with both inflammatory and anti-inflammatory effect. C1 expressing impressive 39.6% of the variance demonstrates a contribution >5% with significant loading factors from inflammatory cytokines including IL-7 (C1L 0.908), TNFα (C1L 0.880), and IL-22 (C1L 0.797) (Figure 1 and Table 2). IL-15 (C1L 0.533) and anti-inflammatory cytokines IL-10 (C1L 0.635) and IL-1RA (C1L 0.643) revealed strong loadings to C1, but contribution < 5% (Table 2). Additional significant contribution was seen from IL-27 (C1L 0.860), also recognized in TNFi responders, having both inflammatory and non-inflammatory properties dependent on interactions with coexistent biomarkers [22]. From the PCA correlation plot (Figure 1), strong associations were found between cytokines IFNγ, IL-10, IL-7, IL-1RA, and IL-8 with the same quadrant positively correlated to both components. They were associated with cytokines TNFα, IL-27, and IL-22 with positive correlation to C1. Interestingly, neither IL-12/IL-23p40 nor IL-17A, associated with PsA immunopathogenesis, contributed significantly to the components. 

### 2.3. Immune Signatures at Baseline in Patients Initiating IL-17Ai

C1 and C2 explained 31.0% and 26.0% of the variance in the data measuring cytokine levels in IL-17Ai responders. The first component was driven by strongly correlated inflammatory cytokines) IL-17A (C1L −0.894), IL-22 (C1L −0.930), and IL-27 (−0.857) that contributed significantly to C1. The cytokines were additional associated and correlated with IL-12/IL-23p40 (C1L −0.693) (Table 3). IL-17A and IL-12/IL-12p40 are both cytokines considered essential to PsA immunopathogenesis as a part of the IL-23/IL-17 inflammatory pathway of PsA (Table 3) [4]. Additional cytokines of C1 included IL-7 (C1L 0.696) and the anti-inflammatory cytokine IL-10 (C1L −0.863); both have been associated with arthritis [23,24]. Further, IL-7 demonstrated statistically significant higher levels at baseline (mean change 4.85 [IQR2.93–8.05], *p* = 0.017) compared to IL-17Ai non-responders (mean change 2.77 [1.60–4.90], *p* = 0.017) (Appendix A) suggesting the inflammatory character of IL-7 that has been associated with Th1 cytokine secretion in joints of patients with arthritis [23]. C2 was driven by inflammatory cytokines IFNγ (C2L 0.578), IL-1α (C2L 0.840), and IL-15 (C2L −0.827) and the anti-inflammatory cytokine IL-1RA (C2L 0.800) working its effect through inhibition of IL-1α and IL-1β (Table 3) [25]. For cytokines with strong contribution to the components, the PCA correlation plot demonstrated overall strong correlation between the individual cytokines IL-17, IL-22, IL-27, and IL10 within the same quadrant under the influence of IL-12/IL-23p40. Interestingly, TNFα, CRP, and IL-6 did not contribute significantly to the components and were inversely correlated to Th17 associated inflammatory markers (Figure 2). 

The two first components of IL-17Ai non-responders at baseline explained 24.2% and 20.0% of the variance in C1 and C2, respectively. C1 was driven by correlated cytokines IL-15 (C1L −0.676), IL-7 (C1L −0.711), IL-22 (C1L −0.757), and inversely correlated TNFα (C1L 0.630), all known for their pro-inflammatory properties in PsA [9]. An additional contribution to C1 comprised CRP (C1L 0.630), angiogenesis associated biomarkers; VEGF-A (C1L −0.782), VEGF-C (C1L−0.736), and bFGF (C1L −0.670), and chemokines; Eotaxin (C1L 0.564) associated with eosinophil recruitment, TARC/CCL17 (C1L −0.676) associated with inflammation, and IP-10/CXCL10 (C1L 0.671) and MDC/CCL22 (C1L −0.509) important to homing of cells to inflamed tissues [7,26]. Additional contributions to C2 included inflammatory cytokines IL-27 (C2L −0.513), IL-1α (C2L 0.534), and IL-7 (C2L −0.532), anti-inflammatory cytokine IL-1RA (C2L −0.533), and further, inflammatory chemokines MCP-1 (C2L −0.763) and MCP-4 (C2L −0.847). Interestingly, neither IL-12/IL-23p40 nor IL-17A contributed to the components in IL-17Ai non-responders compared to IL-17Ai responders, where both were strongly correlated to C1 (Figure 2).

### 2.4. Biomarker Change after Four Months in TNFi DAPSA50 Responders and Non-Responders

In the PCA summarizing biomarker contribution to the immune mechanisms of TNFi responders, C1 explained an impressive 36.8%, while C2 explained 16.0% of the variance. Continuous contribution was seen from inflammatory cytokines IFNγ (C1L 0.704), IL-27 (C1L 0.662), and IL-6 (C1L 0.610), indicating the ongoing importance of these cytokines to the cytokine signature of TNFi responders (Table 2). This is supported by the addition of TNFα (C1L 0.651) as an important contributing cytokine to C1. Thus, with contribution < 5% to the component, implied an increased significance to the immune response after TNFi. Comparison of baseline and four month follow up further implied the increasing importance of anti-inflammatory mechanisms possibly driven by IL-10 (CL1 −0.542) or IL-RA (CL1 0.704). Though, only IL-10 was inversely correlated to the remainder. Overall, IFNγ, IL-27, IL-6, IL-1RA fell within the same quadrant closely associated to TNFα and CRP (Figure 1). In TNFi non-responders, C1 explained 31.7% of the variance, while C2 explained 19.2%. As for TNFi responders, the continuous importance of IL-7 (C1L 0.817), TNFα (C1L 0.570), IL-22 (C1L 0.718), and IL-27 (C1L 0.581) (Table 2) as a part of PsA immunopathogenesis in TNFi non-responders was evident in C1 and with the contribution of IFNγ (C2L 0.758) to C2. Additionally, an important inversely correlated contribution of the inflammatory cytokine IL-17A (CC −0.839) was demonstrated and IL-17A had evidently become a significant contribution to the immune signature of TNFi non-responders compared to the baseline contribution. Further, significant contribution from IL-1α (C1L −0.761) was added to C1, and from IL-12/IL-23p40 (C2L 0.597) to C2 (Table 2). However, IL-12/IL-23p40 loadings were inversely correlated to IL-17A in both components, possibly suggesting alternative pathways from the classical IL-23/IL-17 pathway associated with increasing importance of IL-17A. Overall, IL-17A and IL-1α were inversely correlated to most other cytokines of significance—TNFα, IL-27, and IFNγ—whereas both clusters were associated with IL-22 and IL-7 (Figure 1). 

### 2.5. Biomarker Change after Four Months in IL17Ai DAPSA50 Responders and Non-Responders

The PCA evaluating the immune signature in IL-17Ai responders at four month follow-up revealed that C1 explained 42.7% of the variance, while C2 explained 20.1% of the variance. Main drivers of C1 included Th1 cytokine IFNγ (C1L 0.861) and the induced cytokines IP-10/CXCL10 (C1L 0.935); the latter did not contribute significantly to the components at baseline, and neither did IL-8 (C1L 0.841). With C1 explaining twice as much as C2, IL-12/IL-23p40 (C2L: 0.589) and IL-17i (C2L: 0.755) are considered less prominent cytokines within the immune signature at four month follow-up compared to baseline suggesting a switch in the immune signature in IL-17Ai responders. The PCA evaluating biomarkers at four month follow-up in non-responders to IL-17Ai revealed C1 explaining 22.4% of the variance, while C2 explained 19.4% of the variance. Significant contribution to the C1 was provided by correlated inflammatory cytokines IFNγ (C1L −0.534), IL-12/IL-23p40 (C1L −0.712), and IL-8 (C1L −0.632). All three were not considered significant cytokines of the components at baseline. IL-12/IL-23p40 and IFNγ were additionally strongly correlated with TNFα (C1L −0.616) and within the same quadrant of the correlation plot (Figure 2). Results imply a possible increasing importance of a Th1 and Th17 inflammatory response to the immune signature in DAPSA50 non-responders to IL17Ai. Additional changes to the biomarker signature at follow up included less significant contribution from inflammatory products IP-10 (C1L 0.143/C2L 0.003) and CRP (C1L 0.068/C2L 0.000) and the anti-inflammatory cytokine IL-1RA (C1L 0.273/C2L −0.209).

### 2.6. Baseline Characteristics of MTX Initiators

Ancillary analysis evaluating the baseline cytokine signature of PsA patients initiating treatment with MTX (Appendix A) was conducted to provide an analysis of cytokine signatures in a population that was close to treatment and bDMARD naïve, i.e., a baseline immune signature not subjected to strong immune modulatory therapies, as the study aimed to analyze inflammatory markers in blood plasma obtained before starting the medical therapy. C1 and C2 explained 28.9% and 17.5% of the total variance. C1 was driven by the strongest contribution from correlated cytokines TNFα (C1L 00849), IL-8 (0.928), IL-7 (C1L 0.975), and additionally with strong contribution from CRP (CL1 0.946). C2 was driven by correlated cytokines IL-27 (C2L 0.781) and IL-22 (C2L 0.604). While the contribution and close association between cytokines IL-22, IL-27, and IL-7 was a continuous pattern in all described cytokine signatures, the contribution and relationship between additional cytokines was that TNFα and IL-8 were most similar to TNFi initiators. However, with only little influence of IFNγ on both C1 and C2 of MTX initiators. Considering clinical parameters, the PsA patients constituting these subgroups were also comparable, with the majority being bDMARD naïve and with a tendency of shorter disease duration.

## 3. Discussion

The study aimed to characterize baseline cytokine signatures and interactions in PsA patients, evaluating possible phenotypic differences between responders and non-responders to bDMARD therapies, TNFi, and IL-17Ai, and assess changes to the signatures over time. The PCA was implemented as an established tool to examine phenotypic traits found in the variance of retrieved data of the patient subgroups [27,28]. Homogeneous clinical phenotypes (Table 1) justified the comparison of patients on biomarker level. 

Cytokines associated with PsA immunopathogenesis [29] demonstrated a significant contribution to individual PsA patient subgroups. Thus, phenotypic traits were revealed comparing DAPSA50 responders and non-responders of TNFi and IL-17Ai. It applied to all PsA patient subgroups, TNFi responders, TNFi non-responders, IL-17Ai responders, IL-17Ai non-responders, and MTX initiators that cytokines, IL-7 and IL-27, demonstrated a significant contribution to the cytokine signatures. The same pattern applied to IL-22. Though, IL-22 was not a significant contributor to the cytokine signature of TNFi responders. IL-7 plays a central part in T cell maturation and survival [30], providing a foundation for an increased number of specific T cell subtypes known to PsA. IL-7 has been found to effectively stimulate the production of IFNγ and TNFα in rheumatoid arthritis [23] and IL-17 in spondyloarthritis [31]. In the majority of PsA patient subgroups, IL-22 was strongly correlated with IL-27. IL-22 is considered a pro-inflammatory cytokine in the immunopathogenesis of PsA [9,32], while IL-27 is considered a polyfunctional cytokine with inflammatory properties affecting a wide range of cells, and anti-inflammatory properties promoting IL-10 [22]. Interestingly, IL-27 in both TNFi non-responders, IL-17Ai initiators, and MTX initiators was associated and correlated with IL-22, IL-17A, and/or TNFα at baseline suggesting an inflammatory influence of IL-27 in PsA patients supported by a previous review on IL-27 in a PsA patient cohort [33]. 

Examining phenotypic differences indicated the importance of the Th17 inflammatory pathway, including IL-12/IL-23/IL-17 signaling. However, the revealed fascinating traits possibly explain why IL-17Ai non-responders did not achieve the DAPSA50 treatment response. Mechanisms including TGFβ and IL-6 [34] in association with IL-23 [35] are well-known inducers of Th17 cells, in which research indicates that IL-23 might be responsible for the pathological effect of Th17 leading to IL-17A production and inflammation [36]. Both IL-12/IL23p40, IL-17A, and IL-6 did not contribute significantly to the immune signature of IL-17Ai non-responders (Table 3). On the contrary, the study demonstrated the close association between IL-12/IL-23p40 and IL-17A in TNFi and IL-17Ai responders; although, only with strong contribution to the components in IL-17Ai responders. These results suggest differences in cytokine phenotypes and possibly the need for different medicinal therapies in individual PsA patient subgroups. 

Despite this, Th17 associated inflammatory mechanisms are the main postulated drivers of PsA. Results from this study further implied the contribution from Th1 associated inflammation as well with closely correlated biomarkers TNFα and CRP, and IFNγ together with IL-12/IL-23p40 [37,38] in MTX initiators, TNFi and IL-17Ai responders. IFNγ is known to promote Th1 cell differentiation and inhibit Th17 cell differentiation [39,40], indicating the continued importance of the Th1 inflammatory pathway in PsA immunopathogenesis. This is supported by recent evidence from our group implicating different immune cellular phenotypes in PsA, which additionally suggested that the Th1 driven phenotype was associated with shorter disease duration [41]. This is in line with the current study revealing a strong contribution of TNFα and/or IFNγ, together with only little contribution from IL-12/IL-23p40 and IL-17A in MTX and TNFi initiators, i.e., PsA patient subgroups with a tendency of having shorter disease duration (Table 1) and treated with less bDMARD prior to the current treatment initiation. However, the significant contribution from IFNγ to all cytokine signatures (except IL-17i non-responders) may be attributed to T cell plasticity in which Th17 cells developed IFNγ-secreting properties [42]. These results suggest the importance of dynamic T cell functionality in PsA, which has also been proposed in previous research retrieving both Th1 and Th17 cells from the arthritic joint [43].

Comparing differences between baseline levels of biomarkers revealed only small differences and only between IL-17Ai responders and IL-17Ai non-responders. Besides the cytokines, IL-7, IL-10, and IL-8, already discussed, increased levels of VEGF-A, VEGF-C, MIP-1β, TARC, MIP-3α, and CRP in IL-17Ai responders were reported at baseline possibly mirroring increased inflammation in IL-17Ai responders. However, it is important to note that both angiogenesis associated biomarkers, VEGF-A and VEGF-C, the chemokine TARC, and CRP were all significant contributors to the inflammatory signature of IL-17Ai non-responders as well, indicating an overall influence of the biomarkers to the signatures of PsA patients initiating IL-17Ai.

Data from the four-month follow-up visits confirmed changes to the cytokine signatures, which differed depending on the response to treatment. The continued substantial contribution of inflammatory cytokines might be explained by the DAPSA50 stratification as it is most likely that the subgroup of PsA patients improving ~50% in DAPSA still have an on-going inflammatory response. Studies examining changes in cytokines, TNFα, IFNγ, IL-21, IL-6, IL-8, IL-10, and IL-17 during TNFi demonstrated an initial increase in cytokine levels [44,45,46] further, supporting the sustained contribution to immune signature revealed in the current study. Studies further imply the importance of considering whether four months is sufficient time to conduct a follow-up examination of true changes to cytokine levels due to varying levels of cytokines [44]. 

Limitations of the study include the definition of responders and non-responders based on DAPSA50 corresponding to minor improvement in disease activity [47]. It is assumed that the subgroups of PsA patients that did not reach minimal or low disease activity presented with an ongoing inflammatory process that is reflected in the cytokines measured. It is believed that inclusion of PsA patients reaching DAPSA70 or DAPSA80 would make phenotypic differences clearer than presented in the current study. Four-months follow up might be acknowledged as a limitation considering if four months are sufficient to evaluate changes in the clinical manifestations and the immune response. Four-month follow up was decided in accordance with international recommendations endorsing follow up after 3–6 months as this is the time where a clinical response would be expected or change of treatment would be considered [48].

An additional limitation is that the PsA patients in the study were compared independently of whether receiving bDMARD as monotherapy or in combination with additional csDMARD, and only 31.6% of included IL-17Ai initiators were bDMARD naïve. Previous bDMARD treatment and concomitant csDMARD treatment most likely had an immune-modulatory effect compared to the bio-naïve patients and the patients initiating bDMARD monotherapy, respectively. However, considering individualized treatment of PsA, results are still considered of great interest as patients initiating IL-17Ai mirror real-life difficult-to-treat patients with high PsA disease activity no different to MTX and TNFi initiators, and the PsA patients in combination therapy still exhibit an inflammatory response associated with high disease activity. Additional examination of the influence of csDMARDs on the inflammatory immune response might be needed. 

Moreover, the study only includes an examination of PsA patients initiating TNFi and IL-17Ai. It is important to acknowledge that several new medical drugs targeting other parts of the inflammatory immune pathway of PsA haven been approved for treatment of PsA, including IL-17A/F inhibitors, JAK inhibitors, IL-23 inhibitors, etc. We believe that it is necessary to study the influence of the individual drugs with different targets as it is expected that they will modulate the immune response differently. Immune modulation induced by the individual drugs is important to understand in order to provide individualized treatment of PsA. Lastly, the PCA is limited in that it does not include any value change in cytokine levels, but only describes cytokine interactions. Further, it was decided to use the two-component PCA not considering additional statistical approaches defining the number of components, which might require inclusion of additional components describing cytokine signatures of individual PsA patient subgroups. However, including the two components explaining the greatest degree of variance secured the comparability between subgroups. 

## 4. Materials and Methods

### 4.1. Study Setup and Participants

Plasma samples were included from a consecutive clinical PsA patient cohort at the Parker Institute, Frederiksberg, Denmark [49], registered on clinicaltrials.gov (NCT02572700). Plasma samples and clinical data were obtained from 49 PsA patients with planned initiation of either TNFi (*n* = 20, 19 starting Adalimumab, 1 starting Certolizumab pegol), IL-17Ai (*n* = 19, 15 starting Ixekizumab, 4 starting Secukinumab), and MTX (*n* = 10). An additional inclusion criterium was established for the current study requiring paired biomarker data, including baseline and four-month follow-up data with a maximum of 10% of missing data for the individual biomarker to be included in the PCA. 

### 4.2. Clinical Examination and Preparation of Blood

PsA patients participated in a baseline visit at treatment initiation and after four months of treatment. Clinical data collected comprised SJC, TJC, SPARCC enthesitis score, US assessing synovial hypertrophy, and PASI. Additionally, patient-reported visual analogue scale (VAS) global health was obtained to retrieve composite measure DAPSA quantifying disease activity. Peripheral blood was collected in EDTA vacutainer tubes (Greiner bio-one, Kremsmünster, Germany) and centrifuged to obtain plasma. Plasma was transferred and stored at −80 °C until analysis.

### 4.3. Analysis of Plasma Markers

The MSD V-Plex 54-plex kit (Mesoscale Discovery (MSD), Meso Scale Diagnostic, LCC, Rocksville, DC, USA) (Appendix A). Thawed plasma samples were diluted in-plate in line with manufacturer’s protocol except for panels, Chemokine panel 1, Cytokine panel 2, and the Th17 panel, which were diluted 2× established in an initial pilot. Plates were read on the MSD QuickPlex SQ 120 with software Discovery Workbench version 4.0.12. All samples were loaded in technical duplicates and measurements with intra-assay coefficient of variation (CV) ≥ 30 were excluded prior to data analysis. Biomarker absolute values were given in pg/mL.

### 4.4. Statistical Analysis

Based on exclusion criteria presented in Section 4.1 and Section 4.3, 32 biomarkers (Table 1 and Table 2) were included for the primary analysis. PsA patients were grouped based on the DAPSA50 response (yes/no), i.e., if patients reached a 50% improvement in DAPSA from baseline to follow up. Patient characteristics and biomarker level at baseline were reported as medians with corresponding interquartile ranges (IQR) for continuous variables and number with corresponding percentages for categorical variables. Differences between DAPSA50 responders and DAPSA50 non-responders were evaluated with Mann–Whitney U test and Chi squared test for continuous and categorical variables, respectively. Differences between treatment groups, TNFi, IL-17Ai, and MTX initiators, were evaluated with Kruskal–Wallis test. A two-component PCA was implemented for dimensional reduction to analysis biomarker signatures before and after treatment [50] examining biomarker signatures and the dynamic interactions between the 32 biomarkers. Prior to the PCA, missing data were imputed by the mean of the individual included biomarker variables. PCA components including corresponding eigenvalues were constructed to analyze variance within the dataset. C1 and C2 were retrieved for analysis of the biomarker signature with the largest amount of explained variance in principal component 1 (C1) and the second largest amount of explained variance in principal component 2 (C2). Correlation between individual biomarkers and the components was visualized in correlation plots with the computational configuration of the PCA ensuring that C1 and C2 were uncorrelated [51]. Loadings/correlation coefficients ranging from −1.0 to +1.0 were presented to assess the correlation of the individual biomarker to the components. Strong contribution to the component was defined by the PCA component loading being considered significant at >0.500. Additional subgrouping of results was included defined by contribution percentages > 5%. Statistical analysis was carried out using statistical software R version 4.1.2 with additional relevant packages.

### 4.5. Principal Component Analysis Correlation Plot

The correlation plots (Figure 1 and Figure 2) were included to visualize the relationship between contribution and correlation of the individual biomarkers to the components. Further, to reduce the high dimensional data to easier understandable two-dimensional data. Uncorrelated components; C1 and C2 were represented by perpendicular axes. Degree of contribution was illustrated by colors black to red with black representing minor contribution and red representing strong contribution to the components. Distance between the vector of individual biomarkers demonstrated correlation between biomarkers. Closely associated variables within the correlation plot were positively correlated, while variables in the opposite quadrant were inversely correlated. 

## 5. Conclusions

This study was conducted in an exploratory study setup that provided a broad assessment of the relationship between various cytokines in responders and non-responders to the biological therapies, TNFi and IL-17Ai, which demonstrated different cytokine signatures supporting the hypothesis of different phenotypes leading to diverse immune response mechanisms in PsA [52]. It is believed that interesting knowledge has emerged differentiating responders and non-responders of bDMARDs, which further endorses additional examination of the interactions of the dynamic immune response mechanisms in PsA patients to possibly support future individualized treatment strategies.

## Figures and Tables

**Figure 1 ijms-24-06343-f001:**
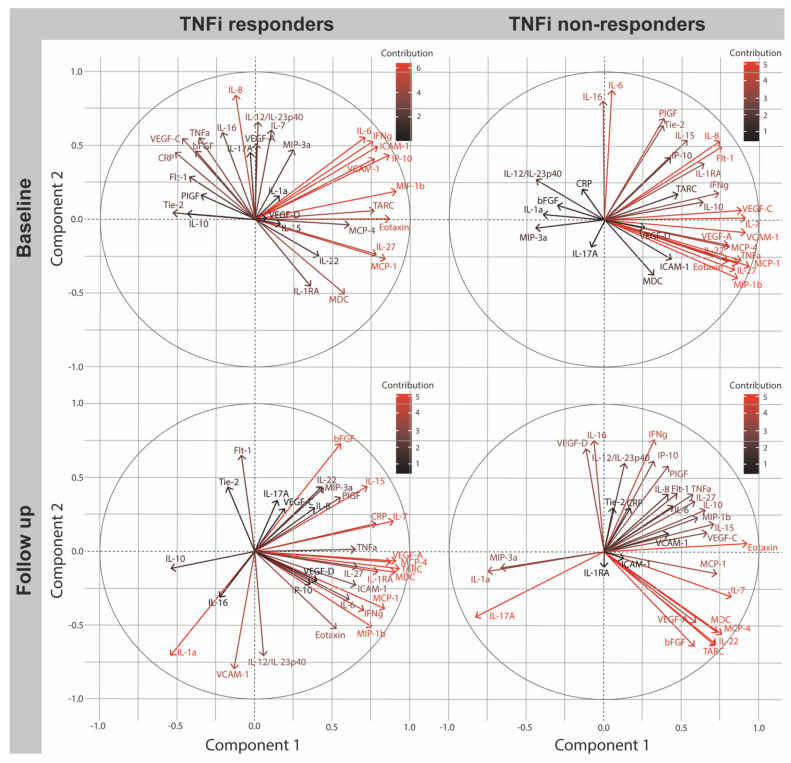
PCA correlation plot including biomarkers of PsA patients initiating TNFi. Correlation plots retrieved from the principal component analysis visualize the relationship between individual biomarker and the components. Component 1 and 2 uncorrelated described with the perpendicular axes from −1.0 to 1.0. The degree of contribution to the immune signature is illustrated with red representing strong contribution to the component and black representing minor contribution. TNFi; Tumor Necrosis Factor-alpha inhibitor, basic Fibroblast Growth Factor, Flt-1; Fms related Receptor Tyrosine Kinase-1, VEGFR1; Vascular Endothelial Growth Factor Receptor 1, PlGF; Placental Growth Factor, Tie-2; endothelial receptor tyrosine kinase, VEGF; Vascular Endothelial Growth Factor, IP-10; IFN-induced protein-10, CXCL; CXC chemokine ligand, MCP; monocyte chemo-attractant protein, CCL; CC chemokine ligand, MDC; macrophage-derived chemokine, MIP; macrophage inflammatory protein, TARC; Thymus and activation regulated chemokine, IL; interleukin, IL-1RA; interleukin 1 receptor antagonist, IFN; interferon, TNF; Tumor Necrosis Factor, CRP; C-reactive protein, ICAM; Intercellular Adhesion Molecule, VCAM; Vascular Cell Adhesion Molecule.

**Figure 2 ijms-24-06343-f002:**
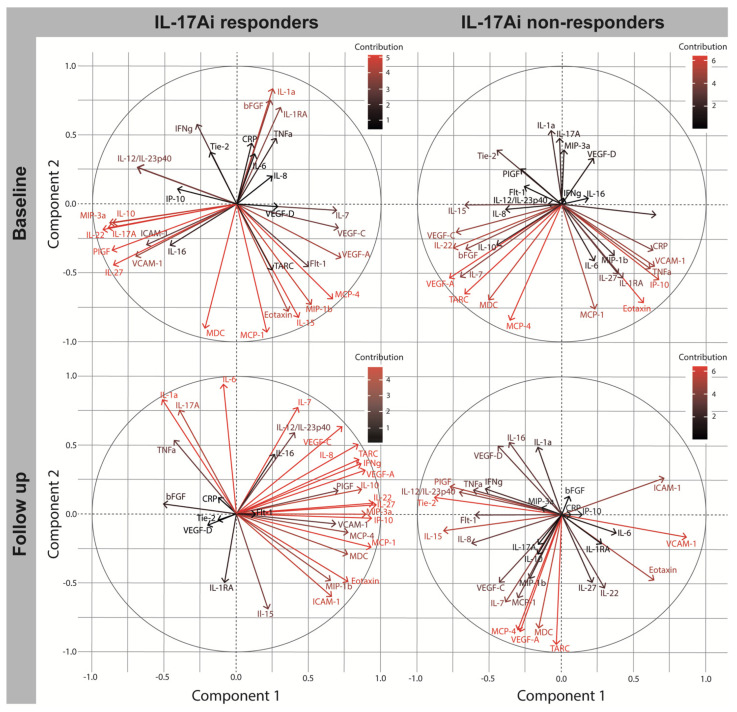
PCA correlation plot including biomarkers of PsA patients initiating IL-17Ai. Correlation plots retrieved from the principal component analysis visualize the relationship between individual biomarker and the components. Component 1 and 2 uncorrelated described with the perpendicular axes from −1.0 to 1.0. The degree of contribution to the immune signature is illustrated with red representing strong contribution to the component and black representing minor contribution. IL-17i; Interleukin 17 inhibitor, basic Fibroblast Growth Factor, Flt-1; Fms related Receptor Tyrosine Kinase-1, VEGFR1; Vascular Endothelial Growth Factor Receptor 1, PlGF; Placental Growth Factor, Tie-2; endothelial receptor tyrosine kinase, VEGF; Vascular Endothelial Growth Factor, IP-10; IFN-induced protein-10, CXCL; CXC chemokine ligand, MCP; monocyte chemoattractant protein, CCL; CC chemokine ligand, MDC; macrophage-derived chemokine, MIP; macrophage inflammatory protein, TARC; Thymus and activation regulated chemokine, IL; interleukin, IL-1RA; interleukin 1 receptor antagonist, IFN; interferon, TNF; Tumor Necrosis Factor, CRP; C-reactive protein, ICAM; Intercellular Adhesion Molecule, VCAM; Vascular Cell Adhesion Molecule.

**Table 1 ijms-24-06343-t001:** Patient characteristics.

	TNFi Responders *n* = 11	TNFi Non-Responders *n* = 9	*p*-Value	IL-17i Responders *n* = 7	IL-17i Non-Responders *n* = 12	*p*-Value	MTX *n* = 10	*p*-Value
Female	7 (63.6)	5 (55.6)	0.714	4 (57.1)	8 (66.7)	0.678	5 (50.0)	0.787 ^†^
Age	48.9 (46.5–55.2)	50.4 (45.5–61.4)	0.470	48.4 (40.7–63.0)	53.5 (45.2–60.8)	0.939	61.0 (54.2–66.0)	0.163 ^†^
Disease duration	4.4 (2.6–7.7)	3.3 (2.1–17.0)	0.970	9.8 (0.9–14.7)	5.9 (2.3–13.7)	0.929	2.3 (1.1–10.6)	0.692 ^†^
Previous bDMARD								
0	8 (72.7)	5 (55.6)	0.437	2 (28.6)	4 (33.3)	0.792	9 (90.0)	**0.049 ^†^**
1	3 (27.3)	2 (22.2)		1 (14.3)	4 (33.3)		1 (10.0)	
2	0 (0.0)	1 (11.1)		2 (28.6)	2 (16.7)		0 (0.0)	
≥3	0 (0.0)	1 (11.1)		2 (28.6)	2 (16.7)		0 (0.0)	
Concomitant csDMARD								
0	9 (0.8)	4 (0.45)	0.134	4 (0.6)	6 (0.5)	0.882	0 (0.0)	**<0.001 ^†^**
MTX	1 (0.1)	4 (0.45)		2 (0.3)	3 (0.3)		0 (0.0)	
SSZ	1 (0.1)	0 (0.0)		0 (0.0)	1 (0.1)		0 (0.0)	
LEF	0 (0.0)	1 (0.1)		1 (0.1)	2 (0.2)		0 (0.0)	
DAPSA	27.0 (22.6–41.3)	38.5 (22.9–39.5)	0.676	37.2 (24.2–38.7)	29.5 (26.2–47.3)	0.708	25.0 (18.1–34.8)	0.208 ^†^
SJC (66)	5.0 (4.5–10.5)	7.0 (3.0–8.0)	0.321	5.0 (4.5–5.5)	6.0 (3.0–11.3)	0.977	6.0 (3.8–8.5)	0.993 ^†^
TJC (68)	7.0 (5.0–12.5)	19.0 (10.0–29.0)	0.303	12.0 (6.0–19.0)	10.0 (8.3–24.0)	0.547	7.5 (5.5–11.0)	0.286 ^†^
SPARCC	3.0 (1.0–6.0)	5.0 (4.0–6.0)	0.514	4.0 (1.5–8.0)	3.0 (1.8–5.0)	0.435	2.5 (0.3–3.0)	0.204 ^†^
US score	8.0 (2.5–8.5)	6.0 (4.0–8.0)	0.508	5.0 (4.5–5.5)	4.5 (1.0–6.3)	0.108	8.0 (5.0–9.0)	0.137 ^†^
PASI	1.5 (0.2–2.5)	1.5 (1.2–5.0)	0.401	1.4 (0.0–2.0)	0.4 (0.0–2.2)	0.930	1.0 (0.7–3.2)	0.416 ^†^

Patient characteristics at baseline were stratified by DAPSA50 response to treatment for the primary analysis. DAPSA50 (yes/no) represents a 50% improvement in the DAPSA from baseline to follow up. Ancillary analysis (†) was conducted including data retrieved from PsA patients before initiation of MTX (unstratified) as 90% was treatment and bDMARD naïve, thereby, only little influence on medical treatment on the clinical outcome and immune response. Patient characteristics were presented as medians with interquartile ranges for continuous variables and numbers with percentages for categorical variables. *p*-values compared TNFi responders versus TNFi non-responders and IL-17Ai responders versus IL-17Ai non-responders. *p*-values of the ancillary analysis (†) compared groups stratified by type of treatment, i.e., TNFi, IL-17Ai, and MTX. Bold numbers represent statistically significant *p*-values < 0.05. DAPSA; Disease Activity in Psoriatic Arthritis, TNFi; Tumour Necrosis Factor-alpha inhibitor, IL-17iA; Interleukin 17 inhibitor, MTX; methotrexate, bDMARD; biological Disease Modifying Anti-Rheumatic Drug, SJC; swollen joint count, TJC; tender joint count, SPARCC; Spondyloarthritis Research Consortium of Canada enthesitis score, US; ultrasound score (synovial hypertrophy), PASI; Psoriasis Area Severity Index.

**Table 2 ijms-24-06343-t002:** Principal component analysis loadings of TNFi initiators.

	Baseline				Follow Up			
	TNFi Responders		TNFi Non-Responders		TNFi Responders		TNFi Non-Responders	
Biomarker	C1	C2	C1	C2	C1	C2	C1	C2
bFGF	−0.382	0.465	−0.308	0.100	**0.552**	**0.730**	**0.574**	**−0.644**
Flt-1	−0.424	0.288	**0.757**	0.499	−0.087	**0.651**	0.464	0.393
PlGF	−0.348	0.168	0.382	**0.690**	**0.554**	0.365	0.410	**0.579**
Tie-2	**−0.532**	0.044	0.378	**0.642**	−0.177	0.438	0.055	0.296
VEGF-A	0.010	**0.516**	**0.796**	−0.171	**0.911**	−0.065	**0.589**	−0.482
VEGF-C	−0.469	**0.549**	**0.883**	0.066	0.192	0.289	**0.662**	0.124
VEGF-D	0.075	0.012	0.259	−0.049	0.403	−0.196	−0.126	**0.694**
Eotaxin	**0.869**	0.002	**0.840**	−0.338	**0.524**	**−0.526**	**0.919**	0.054
IP-10	**0.867**	0.436	0.426	0.424	0.360	−0.224	0.316	**0.614**
MCP-1	**0.842**	−0.272	**0.934**	−0.310	**0.841**	−0.387	**0.722**	−0.152
MCP-4	**0.608**	−0.039	**0.806**	−0.182	**0.933**	−0.121	**0.756**	**−0.563**
MDC	**0.579**	**−0.508**	0.314	−0.373	**0.906**	−0.142	**0.728**	**−0.545**
MIP-1b	**0.912**	0.191	**0.858**	−0.392	**0.754**	**−0.508**	**0.601**	0.231
TARC	**0.769**	0.061	0.473	0.177	**0.878**	−0.073	**0.709**	**−0.638**
IL12/IL23p40	0.020	**0.663**	−0.446	0.273	0.059	**−0.708**	0.127	**0.597**
IL-15	0.166	−0.036	**0.533**	**0.540**	**0.728**	0.441	**0.706**	0.185
IL-16	−0.212	**0.592**	−0.011	**0.806**	−0.227	−0.309	−0.073	**0.750**
IL-1α	0.157	0.162	−0.402	0.039	**−0.543**	**−0.703**	**−0.761**	−0.135
IL-7	0.108	**0.609**	**0.908**	0.013	**0.897**	0.205	**0.817**	−0.309
IL-17A	−0.029	0.455	−0.090	−0.184	0.145	0.347	**−0.839**	−0.442
IL-1RA	0.360	−0.455	**0.643**	0.384	**0.797**	−0.138	−0.006	−0.106
IFNg	**0.761**	**0.531**	**0.736**	0.181	**0.704**	−0.399	0.319	**0.758**
IL-10	−0.437	0.039	**0.635**	0.122	**−0.542**	−0.116	**0.647**	0.280
IL-6	**0.712**	**0.559**	0.044	**0.879**	**0.610**	−0.325	0.442	0.309
IL-8	−0.124	**0.846**	**0.743**	**0.535**	0.388	0.296	0.409	0.387
TNFa	−0.361	**0.555**	**0.880**	−0.274	**0.651**	0.013	**0.570**	0.387
IL-22	0.414	−0.255	**0.797**	−0.281	0.436	0.436	**0.718**	**−0.630**
IL-27	**0.783**	−0.239	**0.860**	−0.294	**0.662**	−0.104	**0.581**	0.338
MIP-3a	0.248	0.474	−0.450	−0.052	0.445	0.429	**−0.675**	−0.114
CRP	**−0.515**	0.456	−0.147	0.210	**0.783**	0.190	0.163	0.296
ICAM-1	**0.791**	0.497	0.432	−0.266	**0.655**	−0.230	0.121	−0.050
VCAM-1	**0.770**	0.413	**0.908**	−0.083	−0.134	**−0.795**	0.411	0.123

Loadings/correlation coefficients for component 1 and component 2 of the principal component analysis grouped by DAPSA50 response to treatment. DAPSA50 (yes/no) define the 50% improvement in DAPSA from baseline to follow-up. Grey markings represent biomarkers with >5% contribution to the component. Bold text represents significant loading factors >0.500. DAPSA; Disease Activity in Psoriatic Arthritis, TNFi; Tumor Necrosis Factor alpha inhibitor, bFGF; basic Fibroblast Growth Factor, Flt-1; Fms related Receptor Tyrosine Kinase-1, VEGFR1; Vascular Endothelial Growth Factor Receptor 1, PlGF; Placental Growth Factor, Tie-2; endothelial receptor tyrosine kinase, VEGF; Vascular Endothelial Growth Factor, IP-10; IFN-induced protein-10, CXCL; CXC chemokine ligand, MCP; monocyte chemoattractant protein, CCL; CC chemokine ligand, MDC; macrophage-derived chemokine, MIP; macrophage inflammatory protein, TARC; Thymus and activation regulated chemokine, IL; interleukin, IL-1RA; interleukin 1 receptor antagonist, IFN; interferon, TNF; Tumour Necrosis Factor, CRP; C-reactive protein, ICAM; Intercellular Adhesion Molecule, VCAM; Vascular Cell Adhesion Molecule.

**Table 3 ijms-24-06343-t003:** Principal component analysis loadings of IL-17Ai initiators.

	Baseline				Follow Up			
	IL-17Ai Responders		IL-17Ai Non-Responders		IL-17Ai Responders		IL-17Ai Non-Responders	
Biomarker	C1	C2	C1	C2	C1	C2	C1	C2
bFGF	0.232	**0.755**	**−0.670**	−0.331	**−0.513**	0.071	0.047	0.137
Flt-1	**0.501**	−0.459	−0.263	0.125	0.139	−0.004	**−0.602**	0.000
PlGF	**−0.867**	−0.338	−0.290	0.254	**0.702**	0.170	**−0.778**	0.209
Tie-2	−0.189	0.381	−0.453	0.391	−0.134	−0.048	**−0.886**	0.129
VEGF-A	**0.724**	−0.396	**−0.782**	**−0.539**	**0.890**	0.318	−0.292	**−0.846**
VEGF-C	**0.708**	−0.178	**−0.736**	−0.206	**0.729**	**0.634**	−0.435	−0.488
VEGF-D	0.288	−0.026	0.216	0.331	−0.205	−0.081	−0.447	**0.503**
Eotaxin	0.349	**−0.779**	**0.564**	**−0.721**	**0.770**	−0.493	**0.636**	−0.472
IP-10	−0.415	0.107	**0.671**	**−0.552**	**0.935**	−0.032	0.143	0.003
MCP-1	0.205	**−0.933**	0.228	**−0.763**	**0.932**	−0.241	−0.307	**−0.599**
MCP-4	**0.665**	**−0.692**	−0.361	**−0.847**	**0.771**	−0.136	−0.308	**−0.836**
MDC	−0.224	**−0.907**	**−0.509**	**−0.670**	**0.769**	−0.292	−0.162	**−0.821**
MIP-1β	**0.516**	**−0.734**	0.361	−0.377	**0.648**	−0.483	−0.224	−0.461
TARC	0.248	**−0.486**	**−0.676**	**−0.657**	**0.844**	0.396	−0.042	**−0.940**
IL12/IL23p40	**−0.693**	0.263	−0.096	0.004	0.400	**0.589**	**−0.712**	0.167
IL-15	0.424	**−0.827**	**−0.676**	−0.010	0.213	**−0.691**	**−0.824**	−0.114
IL-16	−0.464	−0.304	0.183	0.040	0.262	0.433	−0.369	**0.528**
IL-1α	0.249	**0.840**	−0.077	**0.534**	**−0.518**	**0.829**	−0.169	0.493
IL-7	**0.696**	−0.050	**−0.711**	**−0.532**	0.423	**0.772**	−0.393	**−0.631**
IL-17A	**−0.894**	−0.177	−0.018	0.479	−0.399	**0.755**	−0.165	−0.214
IL-1RA	0.302	**0.700**	0.423	**−0.533**	−0.089	**−0.502**	0.273	−0.209
IFNγ	−0.280	**0.578**	0.008	0.041	**0.861**	0.363	**−0.534**	0.193
IL-10	**−0.863**	−0.137	−0.453	−0.298	**0.864**	0.179	−0.172	−0.287
IL-6	0.121	0.364	0.224	−0.414	−0.095	**0.942**	0.375	−0.133
IL-8	0.245	0.120	−0.393	−0.040	**0.841**	**0.505**	**−0.632**	−0.204
TNFα	0.275	0.476	**0.609**	−0.470	−0.436	**0.535**	**−0.616**	0.181
IL-22	**−0.930**	−0.188	**−0.757**	−0.322	**0.951**	0.079	0.292	**−0.530**
IL-27	**−0.857**	−0.448	0.391	**−0.513**	**0.967**	0.065	0.208	−0.490
MIP-3α	**−0.884**	−0.137	0.012	0.396	**0.903**	−0.002	−0.144	0.052
CRP	0.103	0.442	**0.630**	−0.337	−0.130	0.117	0.068	0.000
ICAM-1	**−0.627**	−0.220	**0.664**	−0.053	**0.653**	**−0.599**	**0.707**	0.266
VCAM-1	**−0.703**	−0.385	**0.641**	−0.456	**0.687**	−0.071	**0.863**	−0.161

Loadings/correlation coefficients for component 1 and component 2 of the principal component analysis grouped by DAPSA50 response to treatment. DAPSA50 (yes/no) define the 50% improvement in DAPSA from baseline to follow-up. Grey markings represent biomarkers with >5% contribution to the component. Bold text represents significant loading factors >0.500. DAPSA; Disease Activity in Psoriatic Arthritis, IL-17i; Interleukin 17 inhibitor, bFGF; basic Fibroblast Growth Factor, Flt-1; Fms related Receptor Tyrosine Kinase-1, VEGFR1; Vascular Endothelial Growth Factor Receptor 1, PlGF; Placental Growth Factor, Tie-2; endothelial receptor tyrosine kinase, VEGF; Vascular Endothelial Growth Factor, IP-10; IFN-induced protein-10, CXCL; CXC chemokine ligand, MCP; monocyte chemoattractant protein, CCL; CC chemokine ligand, MDC; macrophage-derived chemokine, MIP; macrophage inflammatory protein, TARC; Thymus and activation regulated chemokine, IL; interleukin, IL-1RA; interleukin 1 receptor antagonist, IFN; interferon, TNF; Tumor Necrosis Factor, CRP; C-reactive protein, ICAM; Intercellular Adhesion Molecule, VCAM; Vascular Cell Adhesion Molecule.

## Data Availability

The data underlying this article cannot be shared publicly due to the privacy of the individuals that participated in the study. Data may be shared as part of research collaborations between participating institutions in line with GDPR and if approved by the Parker Institute and Danish authorities.

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
