# Peer review of "Cytokine Signatures in Psoriatic Arthritis Patients Indicate Different Phenotypic Traits Comparing Responders and Non-Responders of IL-17A and TNFα Inhibitors"

_ijms, 2023, doi:10.3390/ijms24076343_

Round 1

Reviewer 1 Report

1. What is the main question addressed by the research?

It would be better that the authors could clarify this point considering that combination therapy is widely considered the gold standard of treatment.
2. Do you consider the topic original or relevant in the field? Does it
address a specific gap in the field?

The topic is original and relevant in the field regarding therapeutic response to TNF inhibitors or IL-17 inhibitors. 
3. What does it add to the subject area compared with other published
material?

Given that JAK inhibitors such as tofacitinib and upadacitinib have recently been approved for the treatment of PsA, some comments on this issue are desirable since these drugs possess selectivity in blocking different cytokines between responders and non-responders.
4. What specific improvements should the authors consider regarding the
methodology? What further controls should be considered?

No further comment.
5. Are the conclusions consistent with the evidence and arguments presented
and do they address the main question posed?

The conclusions of the revised manuscript are consistent with the available evidence, but some limitations, such as comparison of monotherapy and combination therapy, should be added.
6. Are the references appropriate?

The references seem to be appropriate in the revised manuscript.
7. Please include any additional comments on the tables and figures

No further comment.

Author Response

We thank the reviewer for taking the time to review this manuscript and appreciate comments and recommendations forwarded. We have done our best to address them.

Comments and Suggestions for Authors

  1. What is the main question addressed by the research?

It would be better that the authors could clarify this point considering that combination therapy is widely considered the gold standard of treatment.

Author response: Our main question was to describe the cytokine signature in PsA patients initiating targeted cytokine inhibition therapies TNF inhibitors and IL-17A inhibitors that may explain different response to treatment. We acknowledge the issue that both patients in bDMARD monotherapy and concomitant with csDMARD are compared. However, included patients were initiating bDMARD, TNFi or IL-17i (they did not initiate two therapies as once), due to high disease activity and/or insufficient effect of previous cs/dDMARD, why we might assume that:

1) the immune response at baseline mirrors PsA in high activity and

2) changes in the immune response over time is caused by the effect of the new bDMARD initiated.

In order to clarify, we have

  1. a) made changes to the introduction line 64-70 specifying our goal to investigate the immune response in PsA patients initiating cytokine inhibition therapies, either TNFi or IL-17Ai
  2. b) added information to table 1 to specify number of patients receiving monotherapy or combination therapy, line 99
  3. c) added a section to the discussion further in line with reviewer comment #5 line 389-400
  4. Do you consider the topic original or relevant in the field? Does it address a specific gap in the field?

2. The topic is original and relevant in the field regarding therapeutic response to TNF inhibitors or IL-17 inhibitors.

Author response: We thank reviewer for the comment

  1. What does it add to the subject area compared with other published material?

Given that JAK inhibitors such as tofacitinib and upadacitinib have recently been approved for the treatment of PsA, some comments on this issue are desirable since these drugs possess selectivity in blocking different cytokines between responders and non-responders.

Author response: We thank reviewer for the recommendation and we have added a comment of JAK inhibitors in the introduction (line 56) and the discussion (line 401-408).

  1. What specific improvements should the authors consider regarding the

methodology? What further controls should be considered?

No further comment.

Author response: We thank reviewer for the comment

  1. Are the conclusions consistent with the evidence and arguments presented and do they address the main question posed?

The conclusions of the revised manuscript are consistent with the available evidence, but some limitations, such as comparison of monotherapy and combination therapy, should be added.

Author response: We thank reviewer for the recommendation and have added information to the limitation section of the discussion, line 389-400. Indeed, it would be interesting to evaluate the effect of csDMARDs and csDMARD+bDMARDs on the immune response to describe the difference between patients in monotherapy and combination therapy. Unfortunately, we do not believe that this comparison is possible with the current data available due to low number of patients in combination therapy, i.e., low statistical power.

  1. Are the references appropriate?

The references seem to be appropriate in the revised manuscript.

Author response: We thank reviewer for the comment

  1. Please include any additional comments on the tables and figures

No further comment.

Author response: We thank reviewer for the comment

Reviewer 2 Report

The manuscript is interesting in that it fits into the important line of research of biomarkers for personalized therapy of PsA. However, some points need clarification.

Comments:

1) It appears that patients treated with anti-TNF or anti-IL17A received biologics as monotherapy. The authors should clarify this point considering that combination therapy is widely considered the gold standard of treatment.

2) The authors argue that patients treated with MTX as monotherapy are similar to patients naive to biologics. Since MTX also modulates pro-inflammatory cytokine production they should provide a reference to support their claim. The dosage of MTX used should also be stated where possible.

3) It is unclear the unit of measurement of the various cytokines in plasma and which cytokines component 1 and component 2 contain. This information should be better clarified in the materials and methods section

4) JAK inhibitors such as tofacitinib and upadacitinib have recently been approved for the treatment of PsA. A comment on this is desirable since these drugs possess selectivity in blocking different cytokines.

5) The authors argue that patients treated with anti-TNF were generally already treated with other biologics unlike patients treated with anti-IL17A. If the Danish recommendations indicate that biologics should be preferred as the first to be used, it would be expected that those who received anti-IL17A are the group that received more treatment with biologics. This point needs clarification

6) The follow-up of the study is only 4 months. This is a limitation of the study. A comment on this is needed

7) The analysis of disease activity by MSK US was focused on the presence of synovitis. However, in PsA there are often tendinous, peritendinous, and enthesitis changes detectable by MSK US that are generally considered more disease-specific than synovitis. A comment on this is appreciated

8) Which anti-TNF and anti-IL17A considered in the study should be specified (e.g., adalimumab and secukinumab and/or different molecules?)

9) VEGF, TARC, IL-7, IL-10, MIP-3 and ICAM-1 were found to be statistically correlated with response to anti-IL17A. The authors should discuss the role of these soluble mediators in the pathogenesis of PsA.

Author Response

We thank reviewer for taking the time to review this manuscript and appreciate comments and recommendations forwarded. We have done our best to address them.

Reviewer 2

The manuscript is interesting in that it fits into the important line of research of biomarkers for personalized therapy of PsA. However, some points need clarification.

Author response: We thank reviewer for taking the time to review this manuscript and appreciate comments and recommendations forwarded and have done our best to address them

Comments:

1) It appears that patients treated with anti-TNF or anti-IL17A received biologics as monotherapy. The authors should clarify this point considering that combination therapy is widely considered the gold standard of treatment.

Author response: We thank reviewer for the comment. The majority of included patients received bDMARD as monotherapy. We have added to table 1, line 99, to specify number of patients in monotherapy versus combination therapy. Further, limitations have including both groups of patients have been discussed, line 389-400, also in line with reviewer 1's comments on the subject

2) The authors argue that patients treated with MTX as monotherapy are similar to patients naive to biologics. Since MTX also modulates pro-inflammatory cytokine production they should provide a reference to support their claim. The dosage of MTX used should also be stated where possible.

Author response: We thank reviewer for the comment, but do not entirely agree. We have done our best to edit the manuscript so that the meaning may not be misunderstood. We truly agree with reviewer that MTX modulates the cytokine production. However, what the manuscript intend to was that the baseline cytokine signature of the PsA patients initiating MTX was similar treatment naïve patients, i.e., cytokines have been analyzed before the patients started the MTX. We still believe that this is safe to assume as the analysis of cytokines was performed on blood drawn before the patients started taking MTX. At that time (baseline) the MTX have had no effect on the cytokine production. In accordance with table 1, 9/10 patients did not receive any other drug before MTX, i.e., the data presented in supplementary figure S1 to bDMARD naïve. The single patient that previous received bDMARD did not receive any treatment for >1 year. We have done our best to clarify this in the result section line 79-82 and line 290-294. We did not add any references as we agree with reviewer on the fact that MTX indeed modulates the immune response. Unfortunately, csDAMRD dose was not available.

3) It is unclear the unit of measurement of the various cytokines in plasma and which cytokines component 1 and component 2 contain. This information should be better clarified in the materials and methods section

Author comment: We thank reviewer for the comment. Unit of measurement, absolute value of the biomarker in plasma, has been moved to section 4.3. line 441, instead of section 4.4. As results of the PCA is reported as loadings/correlation coefficients (-1 to +1), they are not presented with units in figure and tables. All 32 cytokines (of 54 measuring with the MSD 54-plex) were included in the principal component analysis based on inclusion/exclusion criteria (section 4.1) and the pre-defined CV cutoff (section 4.3). This has been elaborated, line 443-444.

4) JAK inhibitors such as tofacitinib and upadacitinib have recently been approved for the treatment of PsA. A comment on this is desirable since these drugs possess selectivity in blocking different cytokines.

Author comment: We thank reviewer for the recommendation and have added information of JAK inhibitors to the introduction, line 56, and the discussion, line 401-408.

5) The authors argue that patients treated with anti-TNF were generally already treated with other biologics unlike patients treated with anti-IL17A. If the Danish recommendations indicate that biologics should be preferred as the first to be used, it would be expected that those who received anti-IL17A are the group that received more treatment with biologics. This point needs clarification

Author response: We thank reviewer for the recommendation to clarify. As stated in line 88-89 and reported in table 1, line 99, there is a statistically significant differences in the number of previous bDMARD comparing the established patient groups. As reviewer note, it is expected that patients receiving anti-IL-17A are the group receiving more biologics. Looking at table 1, comparing TNFi initiators only 2 patients have received ≥2 previous bDMARDs, while 8 IL-17Ai initiators have received ≥2 previous bDMARDs (vice versa looking at no previous bDMARDs). We have added a reference to the table, line 89, and clarified that this is cohort specific. The issue has been discussed in line 389-400.

6) The follow-up of the study is only 4 months. This is a limitation of the study. A comment on this is needed

Author response: This was discussed in line 375-377. Follow up was decided based on EULAR recommendations recommending follow up after 3-6 months as we would consider a clinical response at this time point. However, we have added a comment to the discussion section under limitations, line 383-388 and a reference to EULAR recommendations as well.

Gossec L, Smolen J, Ramiro S, De Wit M, Cutolo M, Dougados M, et al. European League Against Rheumatism (EULAR) recommendations for the management of psoriatic arthritis with pharmacological therapies: 2015 update. Annals of the rheumatic diseases. 2015:annrheumdis-2015-208337.

7) The analysis of disease activity by MSK US was focused on the presence of synovitis. However, in PsA there are often tendinous, peritendinous, and enthesitis changes detectable by MSK US that are generally considered more disease-specific than synovitis. A comment on this is appreciated

Author response: Enthesitis was examined by ultrasound in the PIPA cohort from which samples where obtained for the current study. However, for the patients included in the study US findings regarding enthesitis were not as comprehensive and we decided not to include the data for the current study.

8) Which anti-TNF and anti-IL17A considered in the study should be specified (e.g., adalimumab and secukinumab and/or different molecules?)

Author response: We thank reviewer for the comment and have specified the type of medical drug, line 420-421.

9) VEGF, TARC, IL-7, IL-10, MIP-3 and ICAM-1 were found to be statistically correlated with response to anti-IL17A. The authors should discuss the role of these soluble mediators in the pathogenesis of PsA.

Author response: We thank reviewer for the recommendation. We assume that the comment relates to the biomarkers mentioned in line 93-94 that are statistically significant higher in level in IL-17Ai responders compared to non-responders (supplementary table S1). However, this does not necessarily mean that they are correlated to anti-IL17A. During revision we realize that the biomarkers mentioned in the manuscript do not match the supplementary table. The biomarkers not mentioned in the manuscript have been added, line 93-94, which now include VEGF-A (p=0.005), VEGF-C (p=0.022), MIP-1β (p=0.002), TARC (p=0.010), IL-7 (p=0.017), IL-10 (p=0.045), IL-8 (0.004), MIP-3α (p=0.036), and CRP (p=0.009).  Considering the length of the article, we have focused on the discussion of VEGF-A and VEGF-C, MIP-1b, TARC, and MIP-3, and CRP, as the role of IL-7, IL-8 and IL-10 in the pathogenesis of PsA have already been discussed in several section due to the overall contribution to the signatures of the majority of patient subgroups. A section has been added to the result section, line 359-367, it which it is important to note that the majority of mentioned biomarker provide important contribution to the signature of IL-17Ai non-responders as well.

Round 2

Reviewer 2 Report

The authors fully responded to my remarks.